# Electrocatalytic Oxygen Reduction Reaction of Graphene Oxide and Metal-Free Graphene in an Alkaline Medium

**DOI:** 10.3390/nano13081315

**Published:** 2023-04-08

**Authors:** Saravanan Nagappan, Malarkodi Duraivel, SeongHoon Han, Mohammad Yusuf, Manjiri Mahadadalkar, KyeongMun Park, Amarajothi Dhakshinamoorthy, Kandasamy Prabakar, Sungkyun Park, Chang-Sik Ha, Jae-Myung Lee, Kang Hyun Park

**Affiliations:** 1Department of Chemistry, Chemistry Institute for Functional Materials, Pusan National University, 2 Busandaehak-ro 63beon-gil, Geumjeong-gu, Busan 46241, Republic of Korea; yusuf.chem.kor@gmail.com (M.Y.); manjeeri@gmail.com (M.M.); pkmkm951@naver.com (K.P.); 2Department of Electrical Engineering, Pusan National University, 2 Busandaehak-ro 63beon-gil, Geumjeong-gu, Busan 46241, Republic of Korea; malarshalu94@gmail.com (M.D.); prabakar@pusan.ac.kr (K.P.); 3Department of Physics, Pusan National University, 2 Busandaehak-ro 63beon-gil, Geumjeong-gu, Busan 46241, Republic of Korea; han1411@pusan.ac.kr (S.H.); psk@pusan.ac.kr (S.P.); 4School of Chemistry, Madurai Kamaraj University, Palkalai Nagar, Madurai 625021, Tamil Nadu, India; admguru@gmail.com; 5Department of Polymer Science and Engineering, Pusan National University, 2 Busandaehak-ro 63beon-gil, Geumjeong-gu, Busan 46241, Republic of Korea; csha@pusan.ac.kr; 6Department of Naval Architecture and Ocean Engineering, Pusan National University, Busan 46241, Republic of Korea; jaemlee@pusan.ac.kr

**Keywords:** graphene oxide, reduced graphene oxide, graphene, electrocatalyst, oxygen reduction reaction

## Abstract

Graphene is a well-known two-dimensional material with a large surface area and is used for numerous applications in a variety of fields. Metal-free carbon materials such as graphene-based materials are widely used as an electrocatalyst for oxygen reduction reactions (ORRs). Recently, more attention has been paid to developing metal-free graphenes doped with heteroatoms such as N, S, and P as efficient electrocatalysts for ORR. In contrast, we found our prepared graphene from graphene oxide (GO) by the pyrolysis method under a nitrogen atmosphere at 900 °C has shown better ORR activity in aqueous 0.1 M potassium hydroxide solution electrolyte as compared with the electrocatalytic activity of pristine GO. At first, we prepared various graphene by pyrolysis of 50 mg and 100 mg of GO in one to three alumina boats and pyrolyzed the samples under a N_2_ atmosphere at 900 °C. The prepared samples are named G50-1B to 3B and G100-1B and G100-2B. The prepared GO and graphenes were also analyzed under various characterization techniques to confirm their morphology and structural integrity. The obtained results suggest that the ORR electrocatalytic activity of graphene may differ based on the pyrolysis conditions. We found that G100-1B (E_onset_, E_1/2_, J_L_, and *n* values of 0.843, 0.774, 4.558, and 3.76) and G100-2B (E_onset_, E_1/2_, and J_L_ values of 0.837, 0.737, 4.544, and 3.41) displayed better electrocatalytic ORR activity, as did Pt/C electrode (E_onset_, E_1/2_, and J_L_ values of 0.965, 0.864, 5.222, and 3.71, respectively). These results display the wide use of the prepared graphene for ORR and also can be used for fuel cell and metal–air battery applications.

## 1. Introduction

Over the past few decades, oxygen-related electrocatalysis has garnered attention, and much work has been performed to reduce the need for noble metal catalysts or create new, alternative catalysts [1]. To obtain a meaningful output in practical proton exchange membrane fuel cells, platinum has proven to be essential to the oxygen reduction reaction (ORR) [1]. On the other hand, the high cost, limited resources, intolerance to methanol oxidation, and weak durability of Pt-based catalysts hinder their use in fuel cells [2]. Thus, efforts have been made to find cheaper, more active non-precious metal electrocatalysts [3]. Carbon is a potential Pt-alternative catalyst due to its low cost and tenability [4]. ORR electrocatalysts are also very helpful in making metal–air batteries [5,6,7].

Graphite is a cheap carbon-based material that can be readily converted to highly valuable and accessible materials by chemical and thermal treatments [8,9]. Graphene oxide (GO) is typically produced through the oxidative exfoliation of graphite in highly aggressive media [8,9,10]. The properties of the obtained GO are dependent on the graphite precursor, exfoliation method, and purification efficacy. The polar oxygen-containing groups in GO also result in a hydrophilic property, which is advantageous for their easier dispersion in water and exfoliation in a wide range of solvents [10]. In general, the process of preparing graphene involves two distinct phases: first, the oxidation of graphite to produce GO, and then, second, the reduction of GO to produce graphene [11]. GO mainly contains epoxide and hydroxyl groups in addition to trace amounts of carbonyl and carboxyl groups [11]. Graphene is an incredibly fascinating and flexible material that consists of a two-dimensional honeycomb sheet of sp^2^-hybridized carbon atoms [10,11]. Graphene also contains some excellent properties such as a high specific surface area of 2630 m^2^ g^−1^ and electrical conductivity, better thermal conductivity of 5000 W^−1^ K^−1^, optical transmittance (97.7%), Young’s modulus of 1.0 TPa, and intrinsic mobility of 200,000 cm^2^ v^−1^ s^−1^ [11].

Numerous uses for graphene-based materials are found in the fields of energy conversion and storage, construction, health, and the environment. Several routes also exist for obtaining graphenes, such as the exfoliation of graphite, chemical vapor deposition, solvothermal synthesis, the micromechanical cleavage of graphite, and epitaxial growth on an electrically insulating surface [11,12]. To make the production of graphene easier, graphite flakes first undergo an exfoliation process that separates them into individual layers. Chemical and thermal reduction methods were widely used to reduce GO. A variety of toxic reductants, including hydrazine, dimethylhydrazine, hydroquinone, and NaBH_4_, were used in chemical reduction [13]. On the other hand, the thermal reduction of GO is one of the simplest and most promising ways to create significant quantities of reduced graphene oxide (rGO) and graphene which is typically carried out in a vacuum to remove oxygen molecules from carbon π bonds [8,14]. Moreover, this process is much friendlier and easier to reproduce as compared to chemical reduction. In most cases, it has been discovered that the dry GO powder can be decomposed near 210 °C, resulting in a mass loss due to the release of oxygen from the GO [13]. The oxidation of the GO surface can also produce a by-product of oxygen, carbon monoxide, carbon dioxide, and water [8,14]. The oxidation of graphite is accelerated by the presence of functional groups such as hydroxyl and epoxide groups, which decrease the contact between the graphite layers. Epoxy and carboxyl groups are thought to have been removed alongside GO during the thermal reduction process. The reduction of GO into graphene enables the manipulation of characteristics such as electrical and thermal conductivity, surface area, and dispersibility in a variety of solvents [14]. The effect of the thermal reduction temperatures of GO on graphene as well as their property changes due to various temperatures was briefly studied by different research groups [8,14,15]. Mixing a lower amount of graphene (less than 5%) into the polymer would significantly alter the thermal stability, mechanical strength, and flame-retardant properties [15].

Recently, graphene and carbon-based materials were used largely as an electrocatalyst for ORR [16,17,18,19,20]. Higgins et al. briefly reviewed the recent progress and the applicability of graphene and its composites as an electrocatalyst for ORR [20]. The doping of platinum and other noble metals to carbon materials has delivered outstanding electrocatalytic activity for ORR [21,22,23,24]. At the same, the major drawbacks of using Pt-based noble metals are poorer tolerance and stability as well as being expensive [25]. Significant attention is also paid to the introduction of heteroatoms such as nitrogen (N), phosphorous (P), sulfur (S), and boron (B) to the graphene materials, which can reduce the cost of the electrocatalyst and enhance the tolerance and stability against alcohols [16,24,26,27,28,29,30]. Some carbon electrodes have achieved 4e- and 2e- ORR catalytic efficiency comparable to that of these noble metal catalysts under alkaline conditions [5,31,32,33,34]. Ma et al. reported the various possible mechanisms of carbon-based materials for ORR [35]. Kim et al. synthesized N-doped RGO and studied the effect of carbon defects for 2 e- ORR [32]. The authors found that certain carbon defects linked to epoxy or ether groups are more important than nitrogen defects, quinone/catechol groups, or carboxylic acid edge sites when it comes to peroxide formation activity in alkaline conditions. The adsorption of ORR intermediates on carbon materials is thought to be altered in favor of peroxide generation when high-electronegativity heteroatoms are present [33,34,36]. This is because the presence of these atoms activates electrons and induces charge redistribution. On the other hand, Lin et al. reported the 4e- ORR for the N-doped graphene electrocatalyst which is prepared by the pyrolysis of GO with polypyrrole [37]. They investigated the possibility of using polypyrrole (PPy) as the nitrogen source because, at high temperatures, the nitrogen atoms in pyrrole rings can be readily converted into graphitic nitrogen [37]. The use of polypyrrole can induce a larger amount of N-doping (2–3 at%) at the graphene which is responsible for the superior electrocatalytic activity of 4 e- ORR.

In this work, we synthesized GO from graphite by modifying Hummer’s method according to the report method with partial changes in the experimental conditions [38,39]. To obtain GO through a modified version of the Hummers method, potassium persulfate and phosphorus pentoxide are used as graphite pre-treatment reagents unlike the use of sulfuric acid, nitric acid, and potassium permanganate to oxidize graphite by the Hummers method [9,38,39]. Furthermore, we prepared various graphenes by the thermal annealing of the synthesized GO (50 mg and 100 mg of GO in 1 to 3 alumina boats) under a nitrogen atmosphere at 900 °C and naturally cooled it to room temperature. The obtained graphene was used as an electrocatalyst for studying the ORR activity. The main novelty of the work is the direct use of graphene as a four-electron transfer electrocatalyst for ORR. In general, most of the reported carbon samples can show two-electron transfer without the introduction of heteroatoms, transition metals, or noble metals. Our synthesized GO also displayed two-electron transfer pathways during ORR. At the same time, the pyrolysis of GO at 900 °C with the use of 100 mg in one or two boats would almost facilitate the direct four-electron transfer. The four-electron transfer ORR activity of graphene was also confirmed further by preparing various graphenes with similar approaches and finding that the use of an optimum amount of GO during the pyrolysis can facilitate better electrocatalytic ORR activity. We found that without the use of any nitrogen precursor (except nitrogen gas), the electrocatalysts deliver excellent electrocatalytic ORR activity with an electron transfer number of 3.76. Our preliminary results suggest the direct use of graphene as an electrocatalyst for ORR which could be very useful in fuel cell and metal-air battery applications.

## 2. Materials and Methods

### 2.1. Materials

Graphite powder (<20 μm, synthetic), potassium persulfate (K_2_S_2_O_8_, ≥99.0%), phosphorus pentoxide (P_2_O_5_, 99%), potassium hydroxide (KOH) powder, Nafion™ perfluorinated resin solution, and anhydrous ethanol were purchased from Sigma-Aldrich Inc. (Seoul, Republic of Korea). Sulfuric acid (H_2_SO_4_, 97%) and hydrochloric acid (HCl, 35% concentration) was bought from Matsuneon Chemical Ltd. (Osaka, Japan). Potassium permanganate (KMnO_4_, 99%) was supplied by Katayama Chemical Industries Co., Ltd. (Osaka, Japan). Hydrogen peroxide (H_2_O_2_, 30% concentration) was obtained from Junsei Chemical Co., Ltd. (Chuo-Ku, Japan). Double deionized water was used in all experiments. All chemicals were used without further purification.

### 2.2. Synthesis of GO by Modified Hummer’s Method

GO was synthesized by the modified Hummer method according to the reported method with slight changes in the experimental conditions (Appendix A) [9,38,39]. At first, the graphite flake was ground well in a mortar using a pestle followed by dispersing 25 g of the fine ground graphite powder in 1 L of water and ultrasonicating for 30 min. The dispersion was filtered, and the graphite powder was dried overnight at 100 °C in a vacuum oven. Briefly, 10 g of potassium persulfate and 10 g of phosphorus pentoxide were added gradually to 50 mL of concentrated sulfuric acid in a three-neck flask while being stirred until the reagents were dissolved. The pre-treated graphite flakes (15 g) were slowly added into the premixed solution and stirred at 80 °C for 6 h and at room temperature for 12 h [38]. The mixture was slowly infused with 2 L of deionized water while being cooled in an ice bath. The solid was obtained by filtering the liquid suspension through a filter paper and it was then repeatedly washed with deionized water until the pH of the remaining liquid was neutral and subsequently dried at room temperature in a vacuum oven. The oxidized graphite powder was slowly added to a 1 L three-neck flask containing cool condensed sulfuric acid (460 mL) followed by the slow addition of potassium permanganate (60 g) under an ice bath, and the temperature was maintained below 10 °C for 1 h. Deionized water (920 mL) was added slowly under an ice bath, and the suspension was stirred at 35 °C for 2 h. The mixture was transferred to a 5 L beaker followed by the addition of 2.8 L of deionized water. In total, 50 mL of 30% hydrogen peroxide was added slowly to the mixture and the stirring was continued followed by centrifuging the mixture. Wash the sediment further with 10% hydrochloric acid followed by washing it several times with deionized water and drying at 50 °C in a vacuum oven to obtain the GO. The schematic illustration provides a clear route for the synthesis of GO by modifying Hummer’s method.

### 2.3. Preparation of Graphene

We prepared graphene from GO by thermal annealing method. In general, GO has thermal expansion and exfoliation behaviors followed by producing single- or few-layer-thick sheets when the pyrolysis temperature reached above 200 °C. This thermal expansion may limit the bulk production of graphene in a single time. Moreover, this depends on the amount of GO used for pyrolysis and the diameter of the quartz tube as well as the alumina boat dimensions. In this work, we used a standard-size quartz tube fitted for the tubular furnace along with an alumina boat. We studied the effect of the number of alumina boats used for the preparation of a larger quantity of graphene. At first, we placed 50 mg of GO in an alumina boat and pyrolyzed the sample in a tubular furnace at 900 °C for 2 h at a heating rate of 5 °C min^−1^ with a flow of 200 standard cubic centimeters per minute (sccm) of nitrogen (N_2_) (Figure 1). The quartz tube is 70 cm long and 2.5 cm in diameter. In the same way, the alumina boat is 7 cm long and 1 cm wide. The sample was cooled further to room temperature, and the obtained graphene sample was named G50-1B. We also used 2 and 3 alumina boats containing 50 mg of GO in each boat and followed similar experiments condition named G50-2B and G50-3B. The prepared graphene samples almost look like crystalline particles. Likewise, we also took 100 mg of GO in 1 and 2 alumina boats and pyrolyzed them at 900 °C and named them G100-1B and G100-2B.

### 2.4. Characterizations

The surface morphology of the samples was analyzed by the use of field emission scanning electron microscopy (FESEM, SUPRA40, Carl Zeiss AG, Germany) and field emission transmission electron microscopy (FETEM, TALOS F200X, 200 kV, FEI). The samples were evenly put on carbon tape and platinum coated on the substrate before the measurement. The sample was dispersed in ethanol and loaded on a copper grid and dried before measuring with FETEM. An X-ray diffractometer (XPERT-3, Malvern Panalytical Ltd., Malvern, UK) equipped with a Cu anode material and operating at 40 kV in the scanning range of 5–80° 2θ was used to determine the crystalline phase of the samples. Both Fourier transform infrared spectroscopy (FTIR, JASCO (FTIR-4100), Easton, MD, USA) and Raman spectroscopy were able to validate the presence of functional groups in graphene and GO (RAMANtouch). Recordings of FTIR and Raman spectra were made using a scanning range of between 400 and 4000 cm^−1^ and 400 and 3300 cm^−1^, respectively. X-ray photoelectron spectroscopy (XPS) was utilized to acquire information regarding the surface’s chemical state (AXIS Supra, Kratos Analytical Ltd., Manchester, UK). The CASA XPS software was used to perform peak-fitting analyses and quantitative observations. An electrochemical workstation and rotating ring-disk electrode device (RRDE-3A Ver.2.0, ALS Co. Ltd., Tokyo, Japan) with a three-electrode electrochemical cell was used to perform the cyclic voltammetry (CV), linear sweep voltammetry (LSV), and ring-disk electrodes performance at 25 °C. A catalyst sample served as the working electrode while a saturated calomel electrode (SCE) and a graphite rod served as the reference and counter electrodes.

### 2.5. Catalyst Ink Preparation and Electrochemical Measurements

The catalyst ink was created by sonicating 5 mg of catalyst powder with 140 µL of anhydrous ethanol, 140 µL of deionized water, and 20 µL of 5% Nafion solution for 30 min. Before beginning the electrochemical test, the working electrodes (glassy carbon electrode, GCE) of the RRDE were finished with an application of alumina slurry to smooth out any rough edges. After pipetting 7.0 µL of the ink containing a catalyst loading of ~0.117 mg_cat_.cm_disk_^−2^, catalyst inks were dripped over the polished glassy carbon-based rotating ring-disk electrode (RRDE, disk area of 0.1256 cm^2^, and ring area of 0.1021 cm^2^) and allowed to dry at 60 °C overnight. After this, the ORR activity and electron selectivity of the catalysts were evaluated. The electrocatalytic ORR activity was measured at a scan rate of 10 mV s^−1^ in a N_2_ or oxygen (O_2_) saturated 0.1 M aqueous KOH solution (pH = 13).

Using the following equation (1), all potentials were calibrated to a reversible hydrogen electrode [24,26].
E_(RHE)_ = E_(SCE)_ + 0.0592 pH + 0.241 (1)

The Koutecky–Levich equation can be used to calculate the number of electron transfers per oxygen molecule involved in oxygen reduction based on Equations (2)–(4) [24,26].
(2)$$\frac{1}{J}=\frac{1}{J_{L}}+\frac{1}{J_{K}}=\frac{1}{J_{K}}+\frac{1}{B\omega^{1/2}}$$
(3)$$B=0.620nFC_{0}{{(D}_{0})}^{2/3}n^{-1/6}$$
(4)$$J_{K}=nFkC_{0}$$
where J and J_K_ are the measured current density and the kinetic current density. ω, F, C_0_, D_0_, and ν are the electrode rotation rate, Faraday constant (96,485 C mol^−1^), bulk oxygen concentration (1.2 × 10^−6^ mol cm^−3^ for 0.1 M KOH solution), oxygen diffusion coefficient (1.9 × 10^−5^ cm^2^ s^−1^), and kinetic viscosity of the electrolyte (0.01 cm^2^ s^−1^ for 0.1 M KOH solution).

## 3. Results and Discussion

### 3.1. Surface Morphological Analysis

#### 3.1.1. Field Emission Scanning Electron Microscopy (FESEM) Analysis

The surface morphology of GO and graphene were shown in Figure 2a–f. The morphology of GO resembles the formation of a wrinkled and flexible layered structure [15,38]. This is due to the formation of a thick layer of aggregated flake-like sheets. The stacking of the aggregate layers is mainly due to the presence of hydrophilic functional groups may have stronger interactions between each sheet of GO (Figure 2a). The thermal annealing of GO at 900 °C may produce thinner and more loosely packed graphene sheets. This loose texture may be advantageous for better dispersion during catalyst ink preparation. We found that increasing the number of alumina boats with constant GO loading (50 mg and 100 mg) would alter the surface morphology of the prepared graphene (Figure 2a–f). The use of 50 mg of GO in one to three alumina boats would produce almost crystalline graphene particles (G50-1b to 3B) after pyrolysis at 900 °C. The catalyst ink prepared from these samples is dispersed during ultrasonication and loaded on the glassy carbon substrate before analyzing the ORR. At the same time, the pyrolysis performed for the samples with 100 mg of GO in each boat may have shown the formation of lightweight graphene sheets (G100-1B and 2B) as compared to the use of 50 mg of GO (crystalline graphene) due to the better exfoliation and thermal expansion of graphene sheets while introducing a larger quantity of GO in a single boat. For comparison, we also checked this behavior by further increasing the GO amount in a single boat which induces faster expansion of GO sheets even when the temperature reaches above 200 to 250 °C and spreads in the glass tube. This leads to the loss of some samples during collection. To avoid sample loss, we used the optimum amounts of GO (100 mg) samples in a boat for better exfoliation and to avoid the loss of more graphene samples after pyrolysis at 900 °C. The catalyst ink prepared with G100 showed good dispersion like Pt/C ink as compared with the ink prepared with G50. Owing to changes in the expansion and surface morphological differences between G50 and G100, we obtained better electrocatalytic ORR activity for G100 as compared with G50. These results suggest that the use of the optimum amount of GO and thermal annealing temperature plays vital roles to obtain graphene sheets without any loss of samples.

#### 3.1.2. Field Emission Transmission Electron Microscopy (FETEM) Analysis

Transmission electron microscopy (TEM) is another useful tool for determining the quality of exfoliated graphene. The FETEM image of a graphene sheet shown in Figure 3a,b was obtained by dropping a small amount of the graphene dispersion on a copper grid, followed by drying under an air dryer. Kovtyukhova et al. reported the formation of flexible, wrinkled sheets of GO with lateral sizes ranging from hundreds to thousands of nanometers from the TEM image [38]. The FETEM images of the prepared graphene also agreed with the presence of flexible and wrinkled graphene sheets with the formation of a few hundred nanometers of graphene films dispersed well during the thermal annealing at 900 °C (Figure 3a) [11]. At the same time, it is clear that the good dispersion of a single or a few layers of graphene sheets is produced in these conditions. The graphene sheets are so thin that they are almost semi-transparent when placed under an electron beam. The high-resolution TEM image also suggests the presence of a porous structure in the graphene sheet which promotes the passage of gaseous species produced by the breakdown of labile functional groups.

### 3.2. Crystallographic Studies

It is well known that pure graphite has a major X-ray diffraction (XRD) peak at 2θ = 26.5° and a subtle peak at 2θ = 54.5° [40]. These peaks were particular to the (002) and (004) planes, and their d-spacing was 3.5 Å and 1.9 Å, respectively [40]. The XRD patterns of the synthesized GO displayed a sharp peak at 2θ ≈ 10.6° (001) and a minor peak at 2θ ≈ 42.6° (100) with d-spacing values of 8.9 Å and 2.13 Å confirming the successful synthesis of GO (Figure 4A(a)) [40]. Moreover, the reduction in the sharp peak of the graphitic peak around 2θ ≈ 15–35° also confirms the synthesis of GO. The introduction of several oxygen-containing groups on the edge of each layer may be the cause of the increased distance between the layers observed in GO as compared to graphite [13]. This increased the distance between the layers. At the same, the prepared graphene showed the complete disappearance of the peak at 2θ ≈ 10.6° (001) and the generation of sharp graphitic peaks at 2θ ≈ 26° (002), indicating the successful and efficient generation of exfoliated graphene sheets (Figure 4A(b–d)) [13]. The 002 peak intensity partially varied based on the loading of 50 mg of GO in 1–3 boats. It is expected that the conversion of GO to graphene will result in structural changes, particularly in the distance between the layers. Likewise, the XRD patterns of the prepared graphene from 100 mg GO in 1–2 boats also confirm the formation of graphene at 900 °C (Figure 4B(e,f)). A decrease in the d-spacing of graphene is evidence that the thermal reduction process brings about a reduction in the inter-planar spacing. These findings indicate that the process of thermal reduction results in the separation of the layers and the removal of oxygen [13].

### 3.3. Surface Functionality Analysis

#### 3.3.1. Fourier Infra-Red Spectroscopy (FTIR)

We further analyzed the FTIR spectra to confirm the successful synthesis of GO and graphene by the thermal annealing method (Figure 5). The main functional groups present in the GO such as carbonyl groups (C=O from carboxylic acids or ketones) and conjugated alkenes (C=C) were confirmed at 1718 cm^−1^ and 1588 cm^−1^, as well as the other functional groups such as hydroxyl (OH, 3416 cm^−1^), methyl (CH_3_, 1390 cm^−1^), epoxy C-O-C (1222 cm^−1^), and C-O (1042 cm^−1^), respectively (Figure 5a) [41,42]. These peaks confirm the successful generation of GO by the modified Hummers method. The thermal annealing of GO would facilitate the decomposition of carboxylic and other functional groups in the GO surface. The FTIR spectra of the prepared graphene indicate the decomposition of those functional groups present in GO under pyrolysis at 900 °C (Figure 5b–f). The peak that appeared at 1620 cm^−1^ for graphene might be due to the presence of adsorbed water from moisture. In addition to this, a broad peak appeared at 1100 cm^−1^, which was most likely caused by the presence of residual C–O groups [43]. The obtained results also agreed with XRD data and confirm the successful synthesis of GO and graphene.

#### 3.3.2. Raman Spectroscopy

Raman spectroscopy is a crucial tool for characterizing carbon materials because it can tell the difference between single-layer and multi-layered graphene, as well as the size and degree of disorder in the crystals [44]. Figure 6 displays the Raman spectra that were obtained for samples of GO and graphene. GO has demonstrated three significant peaks such as D, G, and 2D bands (Figure 6a) [14]. Among the three peaks, GO has two major peaks such as the D and G bands which appeared at 1334 cm^−^^1^ and 1575 cm^−^^1^, and a very weak broad 2D band centered around 2662 cm^−^^1^ also confirms the successful synthesis of GO [6,34]. The D band is caused by imperfections in the sp2 carbon crystal lattice that are linked to sp3 carbons. In the vast majority of instances, the flaws in the structure of graphene are associated with an unusually broad band of the 2D band [14,41]. The intensity ratios between the D and G bands and their values also provide the structural defect information of GO. In most cases, the ratio of identification disorders to involuntary commitments is a statistic that is utilized in the process of determining the level of disorder. This ratio has a connection with the average cluster size sp^2^ that is related to the value in an inversely proportional way [14,41]. The I_D_/I_G_ ratios of GO appeared around 1.07. On the other hand, graphene samples that have been treated with heat treatments cause a shift in the D (1344 cm^−1^) and G (1585 cm^−1^) bands of the graphene’s Raman spectra (Figure 6b,c) [14,41]. The 2D band disappeared for the thermally annealed graphene (Figure 6b,c). The intensity ratios of G50-1B and G100-1B appeared at 1.21 and 1.15. The higher intensity of the D bands in comparison to the G bands suggests that there is a more disordered phase present in graphene, that the size of the sp^2^ domain has shrunk, and that the sp^2^ network was restored [44]. The presence of oxygen atoms in the material causes an increase in the inter-planar distance and causes a change in its vibrational characteristics. The I_D_/I_G_ ratios for graphene were higher than for GO (1.07), which was indicative of a rising trend for G50-1B (1.21), G100-1B (1.15) and also illustrates the growth of sp^2^ clusters and the potential for the creation of further graphitic domains [14,41].

### 3.4. X-ray Photoelectron Spectroscopy (XPS) Analysis

The XPS survey spectra of GO and graphene (G100-1B) showed two major bands at 285 eV and 532 eV which are due to the presence of the C 1S and O 1s functional groups in the samples (Figure 7A(a,b)). The peak fitted C 1s of GO may display four major peaks such as Sp^2^ carbon (C-C), ether (C-O-C), carbonyl carbon (C=O), and carboxylate carbon (O-C=O) at 284.8 eV, 286.82 eV, 288.3 eV, and 289.7 eV, respectively (Figure 7B) [10,14]. These peaks would confirm the presence of hydrocarbon, hydroxyl, and carboxylic functional groups in GO. The graphitic sp^2^ carbon atoms at a binding energy of about 284.8 eV are responsible for the primary peak in GO [44]. Likewise, the peak fitted O 1s of GO may also manifest the presence of the above functional groups such as O-C=O, C=O, C-OH, and C-O-C at 530.97 eV, 531.77 eV, 532.6 eV, and 533.33 eV, respectively (Figure 7C). These fitting peaks also agreed with the reported values of GO and agreed to the effective synthesis of GO [45]. On the other hand, we noticed an increase in the peak intensity of C1S and a decrease in the peak intensity of O1s indicating the decomposition of functional groups in GO during the pyrolysis temperature (Figure 7A(b)). These results were verified further from the peak of fitted C 1s spectra of graphene obtained at 900 °C under the N_2_ atmosphere (Figure 7D). The carbon Sp^2^ peak at 284.6 eV became a much stronger and more intense peak due to the formation of carbon networks. Meanwhile, the other peak intensities decreased drastically at 285.97 eV, 287.46 eV, and 288.86 eV for C-O-C, C=O, and O-C=O, respectively, owing to the decomposition of the functional groups at 900 °C [41]. This outcome also matched the graphene XPS spectrum created by Johra et al. [11]. Graphene’s XPS spectrum only showed the presence of carbon and oxygen, indicating the lack of impurities and verifying that a high-quality material was prepared using this method [11].

### 3.5. Electrocatalytic ORR Analysis

Recently, carbon-based ORR electrocatalysts that don’t rely on any metals have been proven to be viable low-cost alternatives to platinum-based catalysis because of their stability, accessibility, environmental acceptability, and surface properties [2,23,46]. According to reports, oxygen can be reduced to OH^-^ directly through a four-electron transfer pathway, which has a much higher efficiency at converting chemical energy than the two-electron transfer process that first reduces oxygen to peroxide species (HO_2_^−^; an intermediate followed by a further reduction to OH^−^). We performed cyclic voltammetry (CV) and linear sweep voltammetry (LSV) to predict and confirm the preliminary electrocatalytic ORR activity. These methods are very useful and straightforward to provide a clear view of the electrocatalytic activity of the prepared electrocatalysts and their effectiveness in electron transfers.

At first, we carried the CV of the prepared electrocatalysts such as GO, G50-1B, G50-2B, G50-3B, G100-1B, and G100-2B, respectively, at the N_2_-saturated aqueous 0.1 M KOH electrolyte (pH = 13) (Figure 8A). The LSV graphs of these electrocatalysts performed under a N_2_ atmosphere also agreed with the above results (Figure 8C). It is well known that better ORR activity can be observed with a well-defined cathodic peak operating in 0.1 M KOH under an O_2_ atmosphere [47]. We also checked the electrocatalytic ORR activity under an O_2_-saturated aqueous 0.1 M KOH electrolyte. Onset potential (E_onset_) and half-wave potential (E_1/2_) are the two major parameters used to predict the performance of electrocatalysts for ORR. The CV and LSV graphs of GO under the O_2_-saturated 0.1 M KOH electrolyte display the occurrence of a reduction peak as compared to the GO electrocatalysts under a N_2_-saturated electrolyte (Figure 8B,D). This result indicates the activation of ORR electrocatalytic activity under an O_2_ atmosphere. GO has E_onset_, E_1/2_, and limiting current density (J_L_) values of 0.797, 0.664, and 2.254, respectively. The pyrolysis of GO at 900 °C has shown different ORR responses based on the number of boats kept during the thermal annealing process. G50-1B displayed a better ORR response with E_onset_ and E_1/2_ of 0.802 and 0.656 as compared to G50-2B (E_onset_ and E_1/2_ of 0.785 and 0.640) and G50-3B (E_onset_ and E_1/2_ of 0.763 and 0.618) of GO together at the pyrolysis temperature at 900 °C. Likewise, G50-1B (4.797) presented a higher J_L_ value as compared to G50-2B (3.500) and G50-3B (3.019). The use of 50 mg of GO in a single boat in a tubular furnace would facilitate the uniform transfer of heat throughout the alumina boat, and the pyrolysis process also occurred uniformly on the GO, whereas the introduction of two or three boats (each 50 mg of GO) together would reduce the effectiveness of heat dissipation throughout the samples which may slightly alter graphene preparation and their properties so that the ORR performance is also reduced when introducing more than one boat during the pyrolysis process.

The effectiveness of the use of the number boats and sample loading during pyrolysis was also confirmed further by loading 100 mg of GO in one and two boats together and annealing the GO at 900 °C. We noticed that the use of more than two boats with 100 mg loaded in each boat may lead to the faster thermal expansion of GO when the temperature reached above 250 °C as compared to one or two boats. Therefore, we used the optimum GO amount in the preparation of graphene to avoid sample loss due to the expansion followed by the free flow of the sample in the tubular furnace. We could not see the free flow of the graphene sheet even though the thermal annealing temperature reached 900 °C for G50-1B-3B, whereas the partial expansion of the graphene sheet was observed at G100-1b and G100-2B. We further checked the electrocatalytic ORR activity of these samples. We noticed better electrocatalytic ORR activity for G100-1B (E_onset_, E_1/2_, and J_L_ values of 0.843, 0.774, and 4.558, respectively) and G100-2B (E_onset_, E_1/2_, and J_L_ values of 0.837, 0.737, and 4.544, respectively), with a more positive onset potential and stable reduction of oxygen taking place for these electrocatalysts. The electron transfer number was predicted from the LSV carried out at various rotations per minute (400–2500 rpm, orange arrow indicates the LSV at different rpm from 400-2500 rpm) at the O_2_-saturated aqueous 0.1 M KOH electrolyte. The LSV graphs of GO, G50-1B-3B performed at different rpms indicated the three stages of ORR activity in O_2_-saturated aqueous 0.1 M KOH electrolyte which indicates fairly lower performance in ORR activity (Figure 9A–D). At the same time, G100-1B delivered 2 stages of ORR activity which indicates that excellent ORR activity and stable electron transfer occurred for the G100-1B electrocatalyst (Figure 9E). Likewise, G100-2B also has similar ORR activity with significantly less ORR activity as compared to G100-1B (Figure 9F).

The study on the Koutecky–Levich (K-L) plot is the most common method to prove the electron transfer number (*n*) from the LSV graphs performed at different rpms. The slopes of the K-L plots can be used to calculate the number of transferred electrons (*n*) involved in oxygen reduction as shown in Figure 10. The *n* values of GO, GX 1B, GX 2B, GX 3B, GY 1B, and GY 2B at 0.4 V were 1.69, 2.69, 2.54, 0.68, 3.76, and 3.41, respectively. These results proved the superior ORR activity for G100-1B and G100-2B as compared to other electrocatalysts. The G100-1B and G100-2B electrocatalytic activity can suggest that almost a direct four-electron transfer reaction occurred and may help to produce water as an end product. The better ORR activity of the optimized graphenes may be due to the presence of the higher valence orbital energies of the ORR active atom, which increases the adsorption of OOH* and OH* intermediates.

We also performed a comparison study with commercial Pt/C electrocatalysts due to the benchmark electrocatalytic activity of Pt/C electrocatalysts for ORR. The CV and LSV graphs of Pt/C electrocatalysts performed with N_2_- and O_2_-saturated electrolytes display better ORR activity under the O_2_ atmosphere as compared with the N_2_-saturated electrolyte (Appendix A). In addition, the ORR activity was analyzed further from the LSV carried out under the O_2_ atmosphere under various rotation speeds which indicates superior ORR activity with a stable current flow (Appendix A). The E_onset_, E_1/2_, and J_L_ values of Pt/C electrocatalyst at 1600 rpm are 0.965, 0.864, and 5.222, respectively. Furthermore, we also checked the *n* value from the slope of the K-L plot (Appendix A). The estimated *n* value for the Pt/C electrocatalyst is 3.71. These results indicate that the prepared graphene (G100-1B) and Pt/C electrocatalysts have displayed superior ORR with an *n* value close to four, which suggests the direct production of water during the electrocatalytic reaction.

Several research works have disclosed that outstanding ORR catalytic activity can be attributed to the presence of noble metals, nitrogen, or other heteroatom doping in the graphitic basal plane [43]. Nitrogen functions predominately exist on the edges of the N-doped graphene (NG) and are capable of playing supporting roles in the ORR. These peaks may appear at around 401.2 eV after thermal annealing at higher temperatures. Lin et al. also reported that increasing the pyrolysis temperature from 400 to 900 °C for preparing NG may reduce the percentage content of N in the NG [43]. The XPS survey spectrum of graphene shown in Figure 7A(b) suggests the absence of the N-atom at the binding energy of around 401 eV in the graphene. These results suggest the introduction of N_2_ gas during the pyrolysis of GO at 900 °C may show no stronger sorption of N atoms at the graphene basal planes and edges. At the same time, we discussed the better electrocatalytic ORR activity of graphene prepared at 900 °C under a N_2_ atmosphere as compared with GO and rGO prepared at 250 °C may suggest some atomic-scale N atoms may exist on the graphene or that these may create some defects on the graphene which may be responsible for superior ORR activity. Using an FTIR study, it was also difficult to exactly predict the N-doping on graphene due to the presence of C-N, N-H bending, and C=C stretching peaks present at 1187 cm^−^^1^ and 1571 cm^−^^1^ that may overlap with other functional groups as explained above (Figure 5) [47,48]. We also provided some of the best-performing graphene-based electrocatalysts for ORR (Appendix A). Our prepared graphene electrocatalyst has quite a better performance without any doping as compared to the reported work. Therefore, the preliminary results suggest the straightforward use of our prepared graphene electrocatalyst for ORR without the inclusion of any noble metal or heteroatom doping on the graphene.

## 4. Conclusions

In this work, we synthesized GO from graphite by modifying Hummer’s method followed by preparing graphene via the thermal annealing method under a N_2_ atmosphere. We also checked the reproduction of graphene on a large scale by introducing similar amounts of GO in different alumina boats (50 and 100 mg in each boat). We studied the effect of crystallinity and the expansion of GO under thermal annealing temperatures via XRD and also confirmed the surface property and structural behaviors under various characterization techniques. The overall study conveys the successful synthesis of GO and the effective production of graphene based on tubular furnace diameter and annealing temperatures. Our preliminary results hypothesize that the graphene production may differ based on the tubular furnace quartz tube diameter. Based on the tubular furnace quartz tube diameter the use of the alumina boat size also differs, which may affect the graphene production. Furthermore, we studied the electrocatalytic activity of GO and graphenes. We noticed excellent electrocatalytic behaviors of graphene prepared by the introduction of 100 mg of GO in one or two boats (G100-1B and 2B) at the pyrolysis temperature of 900 °C under an O_2_ atmosphere as compared to GO and G50-1B to 3B. We found that G100-1B (E_onset_, E_1/2_, J_L_, and *n* values of 0.843, 0.774, 4.558, and 3.76) and G100-2B (E_onset_, E_1/2_, and J_L_ values of 0.837, 0.737, 4.544, and 3.41) displayed better electrocatalytic ORR activity similarly to Pt/C electrodes (E_onset_, E_1/2_, and J_L_ values of 0.965, 0.864, 5.222, and 3.71, respectively). We strongly believe that the G100-1B electrocatalytic ORR activity (without any doping) is almost closer to the electrocatalytic ORR of commercial Pt/C electrocatalysts. These findings open up wider opportunities and the graphene properties can be easily turned with slight modifications. We also hope that the ORR activity can be enhanced further by the introduction of atomic-scale noble metals or heteroatoms in prepared graphene. More detailed studies need to be performed on the optimized electrocatalyst to improve the activity further for superior electrocatalytic ORR activity. Our initial findings point to the potential utility of graphene as an electrocatalyst for ORR in fuel cells and metal–air batteries.

## Figures and Tables

**Figure 1 nanomaterials-13-01315-f001:**
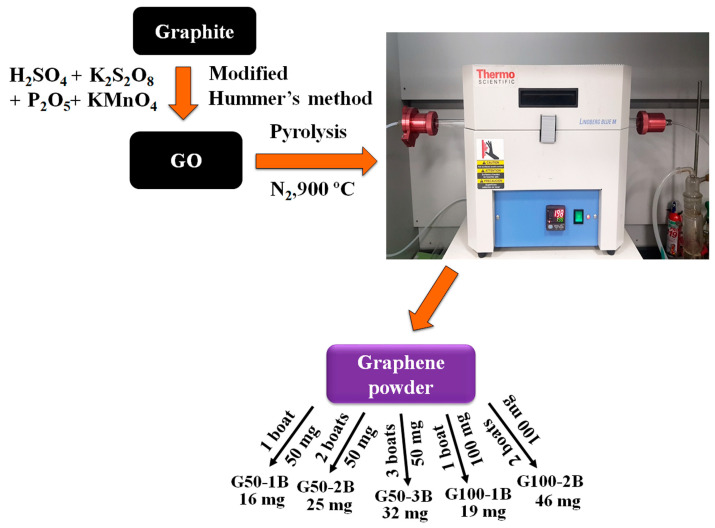
Schematic representation of the preparation of graphene from GO.

**Figure 2 nanomaterials-13-01315-f002:**
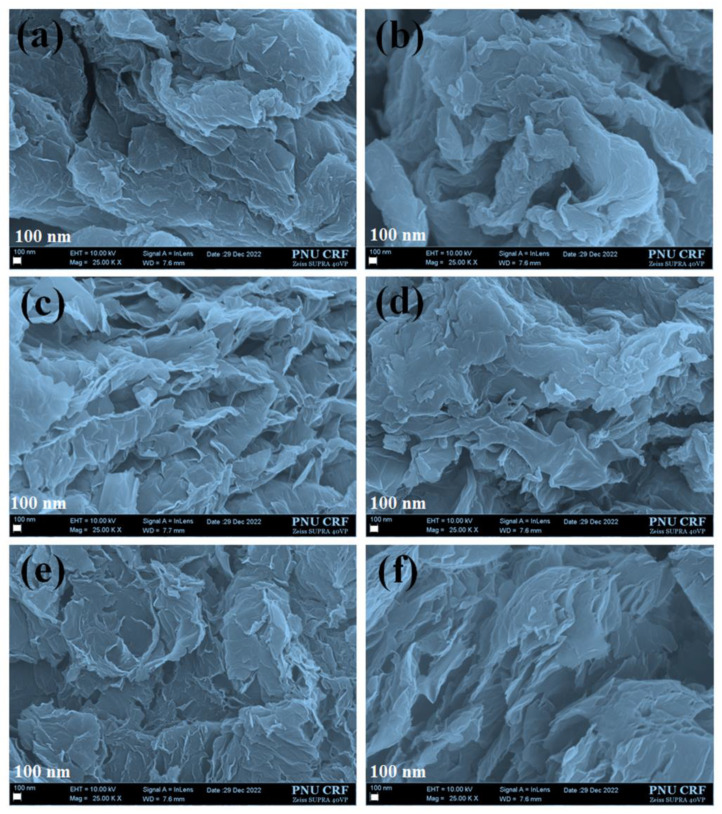
FESEM images of (**a**) GO, (**b**) G50-1B, (**c**) G50-2B, (**d**) G50-3B, (**e**) G100-1B, and (**f**) G100-2B, respectively.

**Figure 3 nanomaterials-13-01315-f003:**
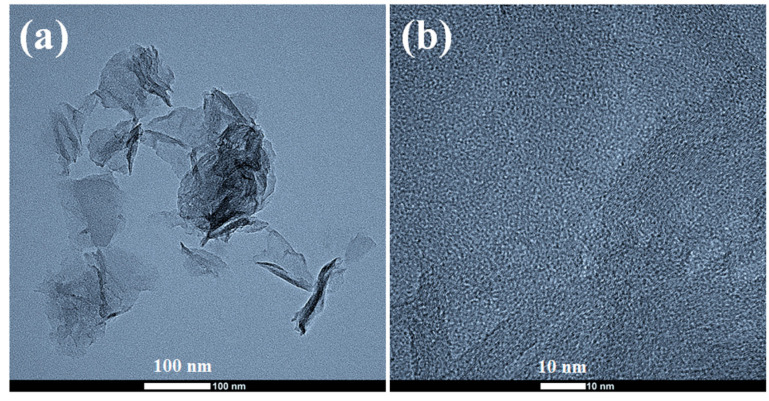
(**a**,**b**) FETEM images of the prepared graphene (G100-1B).

**Figure 4 nanomaterials-13-01315-f004:**
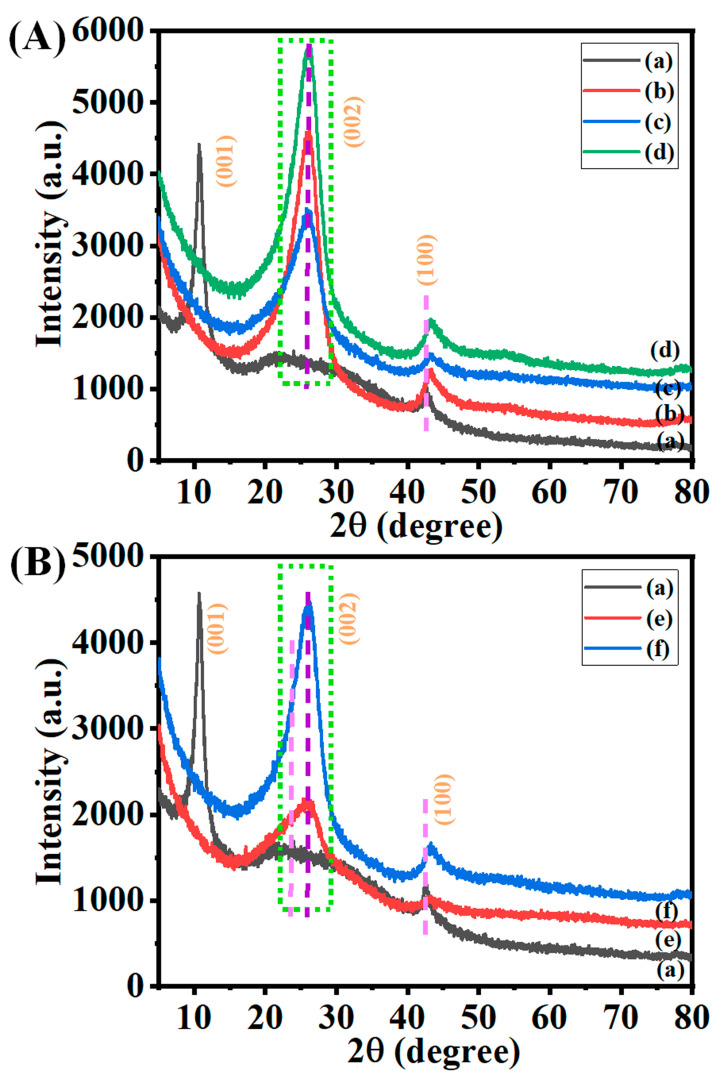
(**A**,**B**) XRD patterns of (**a**) GO, (**b**) G50-1B, (**c**) G50-2B, (**d**) G50-3B, (**e**) G100-1B, and (**f**) G100-2B, respectively.

**Figure 5 nanomaterials-13-01315-f005:**
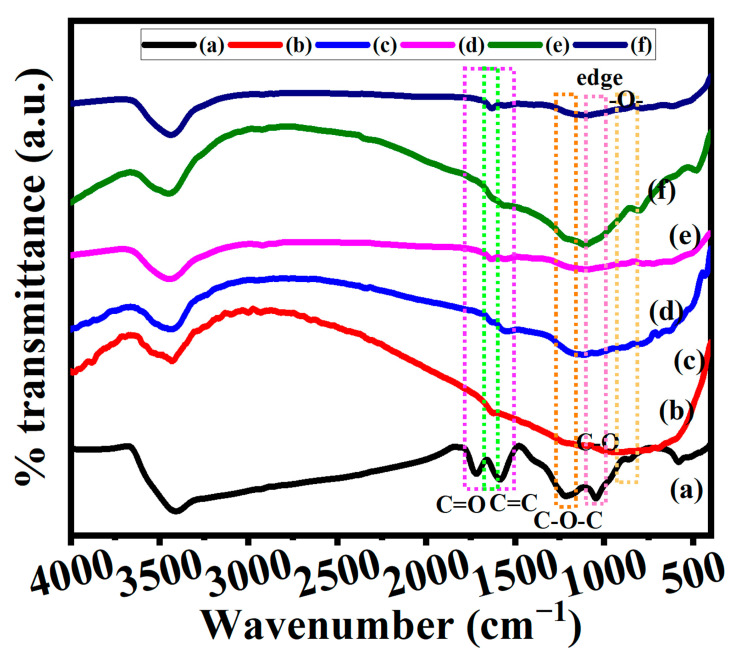
FTIR spectra of (**a**) Go, (**b**) G50-1B, (**c**) G50-2B, (**d**) G50-3B, (**e**) G100-1B, and (**f**) G100-2B, respectively.

**Figure 6 nanomaterials-13-01315-f006:**
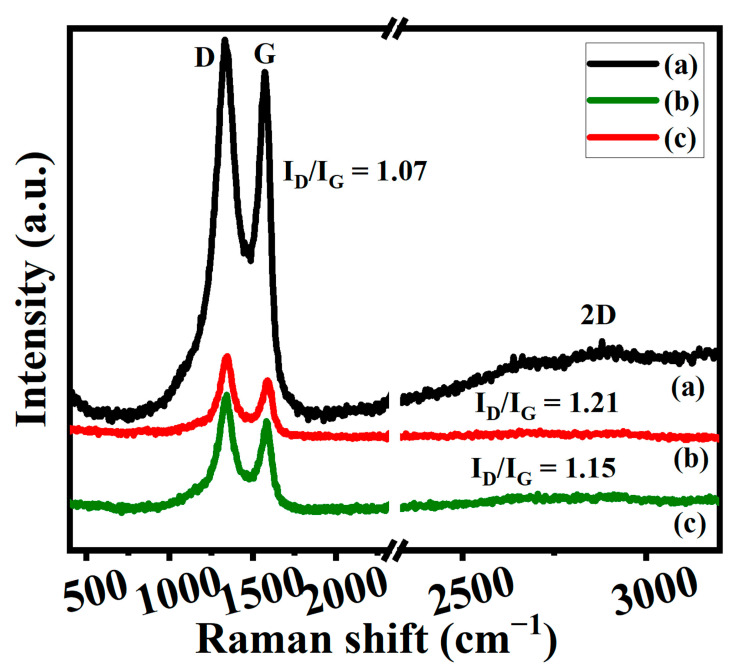
Raman spectra of (**a**) GO, (**b**) G50-1B, and (**c**) G100-1B.

**Figure 7 nanomaterials-13-01315-f007:**
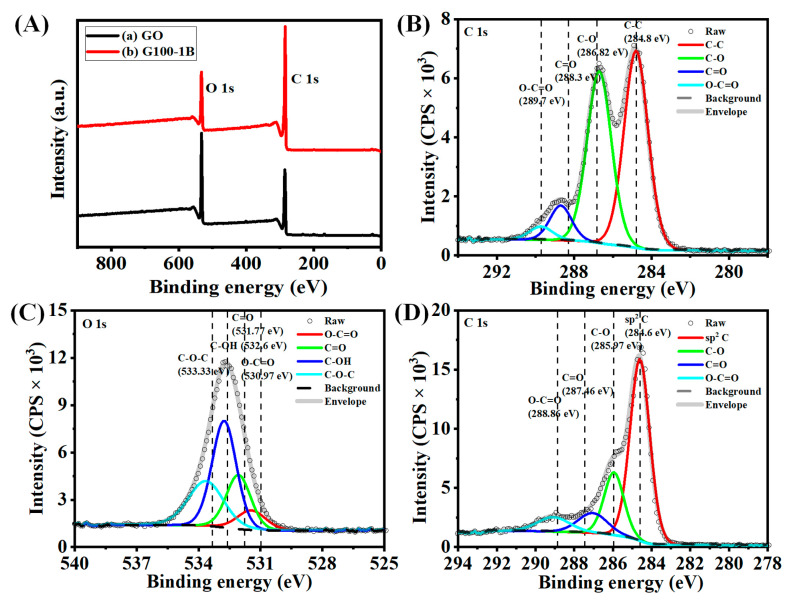
(**A**) XPS survey spectra of (**a**) GO and (**b**) G100-1B. (**B**,**C**) Peak fitted C1s and O1s XPS spectra of graphene oxide (GO), and (**D**) C1s spectra of G100-1B.

**Figure 8 nanomaterials-13-01315-f008:**
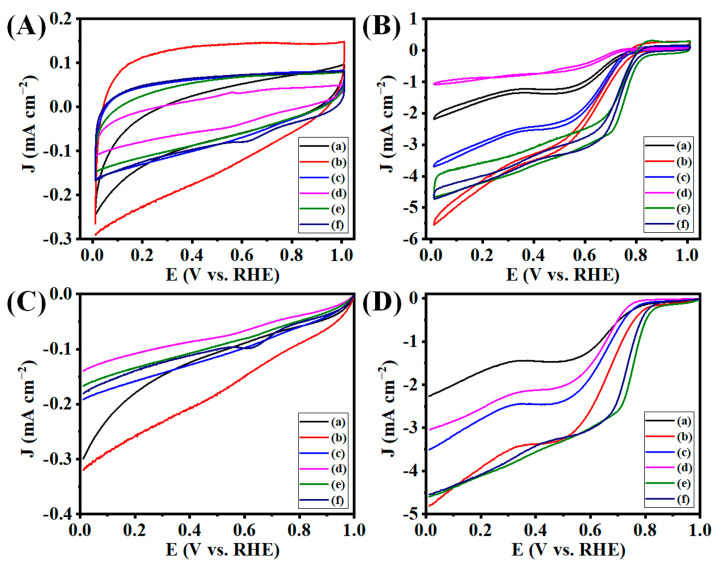
(**A**,**B**) CV and (**C**,**D**) LSV curves of (**a**) Go, (**b**) G50-1B, (**c**) G50-2B, (**d**) G50-3B, (**e**) G100-1B, and (**f**) G100-2B in N_2_- and O_2_ saturated-0.1 M KOH electrolytes at the scan rate of 10 mV s^−1^ with the rotation speed of 1600 rpm at room temperature.

**Figure 9 nanomaterials-13-01315-f009:**
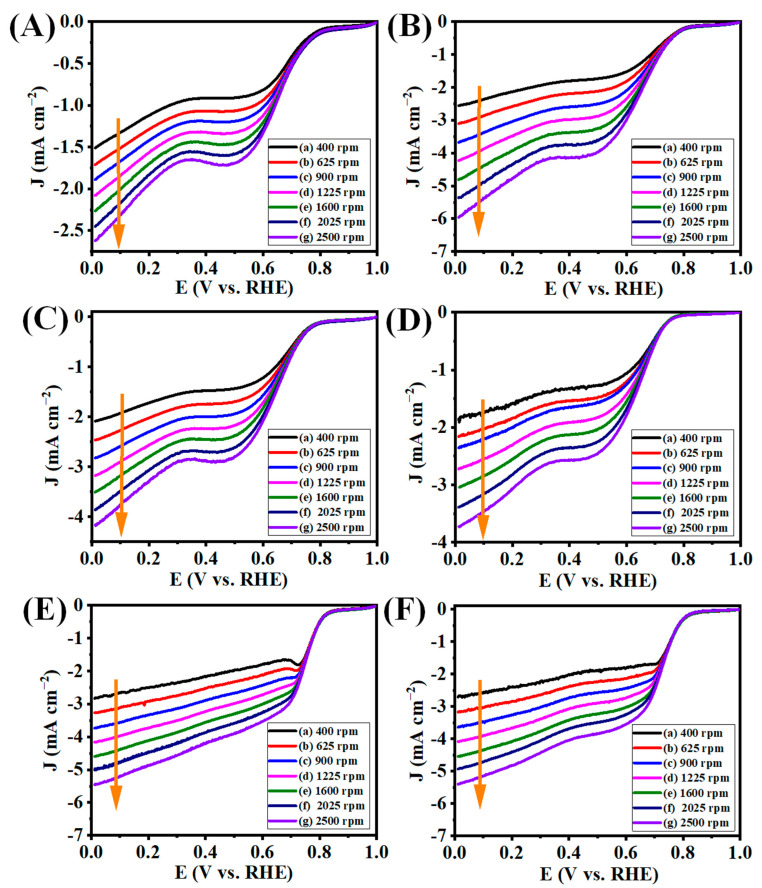
LSV curves of (**A**) GO, (**B**) G50-1B, (**C**) G50-2B, (**D**) G50-3B, (**E**) G100-1B, and (**F**) G100-2B at various rotation speeds in O_2_-saturated 0.1 M KOH electrolyte at the scan rate of 10 mV s^−1^ with different rpm speeds at room temperature.

**Figure 10 nanomaterials-13-01315-f010:**
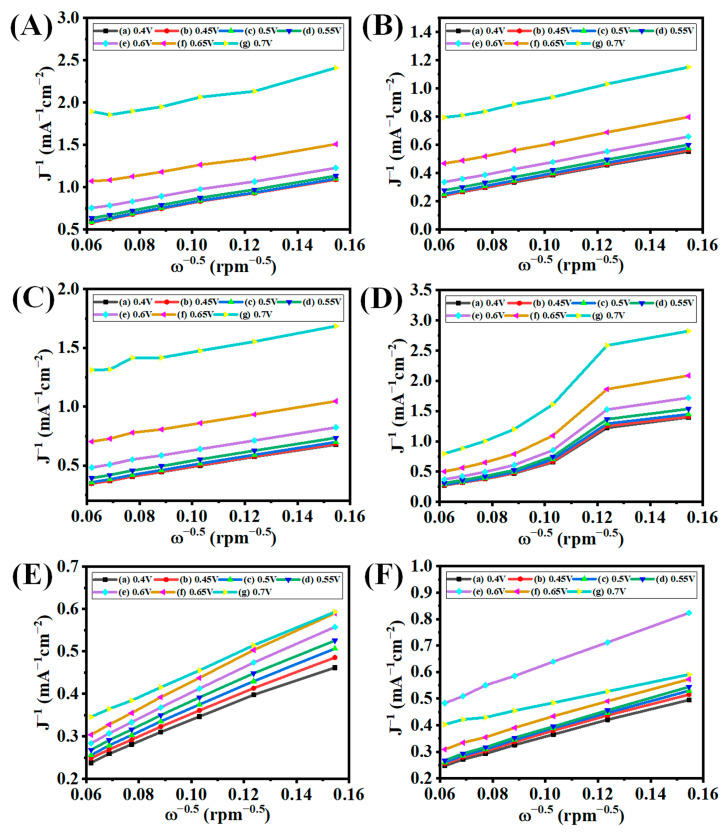
K-L plots of (**A**) GO, (**B**) G50-1B, (**C**) G50-2B, (**D**) G50-3B, (**E**) G100-1B, and (**F**) G100-2B in O_2_-saturated 0.1 M KOH electrolyte at the scan rate of 10 mV s^−1^ with different rpm speeds at room temperature.

## Data Availability

Data are contained within the article.

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
