# Peer review of "Electrocatalytic Oxygen Reduction Reaction of Graphene Oxide and Metal-Free Graphene in an Alkaline Medium"

_nanomaterials, 2023, doi:10.3390/nano13081315_

Round 1

Reviewer 1 Report

Journal: Nanomaterials

Title: Electrocatalytic oxygen reduction reaction of graphene oxide and platinum group metal-free graphene in an alkaline medium

Manuscript ID: 2311438

In the current work, graphene samples were synthesized via thermal treatment of various quantities of graphene oxide (GO) and used alumina boats. The as-prepared samples were tested for the oxygen reduction reaction (ORR) in an alkaline medium, where the graphene synthesized from 100 mg GO and by using 1 alumina boat showed the highest catalytic activity.    

The as-prepared samples were characterized by several physicochemical methods, while the main ORR characterizations and calculations were performed. The effect of GO and alumina boat quantities during the thermal treatment on the physicochemical properties of the as-prepared samples is not adequately revealed. Additionally, the electrochemical characterizations, language, and manuscript structure should be significantly enhanced. Last but not least, the novelty and contribution of the work is not clear. Conclusively, the manuscript is suggested for rejection under the current state.       

1)                  English syntax and grammar need significant improvement. Moreover, there are some typo mistakes in the manuscript (e.g., p.2; line 73, equation 4, etc.). Finally, the text is not well-structured; as a result, in several parts, information is repeated or does not follow the rationale of the preceding information. Conclusively, the entire manuscript should be double-checked and carefully revised.

2)                  The novelty of the work is not clear. The effect of the reduction temperature on the electrocatalytic and physicochemical properties of GO has been widely examined in the past decade, including several works concerning the oxygen reduction reaction. Moreover, recently, investigations have been extended to advanced GO-based samples, indicating the exhaustion of the pure GO examination. What is exactly the novel aspect of the current work relative to similar works previously reported? Is it the examination of various quantities of alumina boats and GO quantities during thermal treatment? Generally, this novelty should be highlighted in the introduction by providing the relevant references and thoroughly discussing them compared to your research. By the way, half of the introduction is devoted to general information about graphene and GO, which has already been reported in several manuscripts. It would be sufficient to focus specifically on the beneficial effect of GO on ORR and on highlighting the novelty. Finally, it is not clear for which applications that include ORR the as-prepared catalyst is suggested.

3)                  A literature review paragraph should be included in the introduction section to highlight the different ways for ORR (last five years). See for example works of prof. Shuqin Song and prof. Zhenxing Liang groups.

4)                  More of the half references are older than 5-6 years. They should be updated.

5)                  The nature and the role of the catalyst active sites for ORR should be better and clearly pointed out. The authors should specifically explain the role by connecting the physicochemical characterization results with the photocatalytic activity of the prepared catalyst.

6)                  In the abstract and introduction sections, numerical ORR activity results should be highlighted. More information about the main findings should be reported in Abstract section.

7)                  The phrase “platinum group-free metal graphene” is not adequate. “Metal-free” would be more sufficient, since the as-prepared graphene is pure. Moreover, the notation X and Y for 50 and 100 mg of GO is complicated. You could note the samples as G50 and G100 instead of GX and GY.

8)                   It is argued that the annealing at 900 oC produces loose and thin graphene sheets that they could assist the catalyst ink dispersion on the working electrode. This argument does not seem correct. How is the microstructure correlated with the ink dispersion? Presumably you were trying to refer to the electrochemical active surface area.  

9)                  The effect of the alumina boats and GO quantity on the physicochemical features of the as-prepared samples are not explained but only demonstrated in the corresponding characterizations.

10)              It is argued that RGOY presents ORR activity in N2 saturated solution. This argument is incorrect. The obtained CV demonstrates the surficial oxide formation/reduction, either because of the reorganization of the catalyst surface or interaction with the electrolyte ions. Moreover, such a CV figure could indicate a not well-organized catalytic surface area, which could also be argued according to the large hysteresis of the CV in the O2-saturated electrolyte. If this is the case, a conditioning procedure is required before conducting ORR measurements for reliable and reproducible results. Indicatively, the kinetic region seems to change slightly with the rpm alternation for the RGOY sample (Figure 9), which should not be the case. Probably the measurements were not reproducible.

11)              For reliable Koutecky-Levich analysis, the ORR LSV curves should be corrected for the ohmic solution resistance and the capacitive (background) currents.

12)              The ORR results analysis should be enhanced, e.g., calculate kinetic currents, exchange current densities, provide durability results, etc.   

13)              According to Table S1 and other reported works, the achieved ORR results are not exceptional. Based on this fact, you should not exaggerate about the contribution of the current work.   

14)              The manuscript is not well-organized. The main ORR results should be transferred from the “Discussion” section to the “Results” section. The “Discussion” section should be exclusively devoted on correlating the electrocatalytic behavior of the prepared samples with the physicochemical results, while highlighting the effect of the alumina boats and GO quantities during the thermal treatment. 

Author Response

Answers to the reviewer's comments

Reviewer 1

In the current work, graphene samples were synthesized via thermal treatment of various quantities of graphene oxide (GO) and used alumina boats. The as-prepared samples were tested for the oxygen reduction reaction (ORR) in an alkaline medium, where the graphene synthesized from 100 mg GO and by using 1 alumina boat showed the highest catalytic activity.    

The as-prepared samples were characterized by several physicochemical methods, while the main ORR characterizations and calculations were performed. The effect of GO and alumina boat quantities during the thermal treatment on the physicochemical properties of the as-prepared samples is not adequately revealed. Additionally, the electrochemical characterizations, language, and manuscript structure should be significantly enhanced. Last but not least, the novelty and contribution of the work are not clear. Conclusively, the manuscript is suggested for rejection under the current state.  

Thank you very much for reviewing our manuscript also for your valuable comments to improve the quality of our manuscript. We carefully revised all your comments and inputted the answer to all your queries point-by-points and also mentioned the changes in the revised manuscript in red colour.         

1) English syntax and grammar need significant improvement. Moreover, there are some typo mistakes in the manuscript (e.g., p.2; line 73, equation 4, etc.). Finally, the text is not well-structured; as a result, in several parts, information is repeated or does not follow the rationale of the preceding information. Conclusively, the entire manuscript should be double-checked and carefully revised.

Response 1: We double-checked the entire manuscript and corrected the grammatical and other errors in the revised manuscript.

2) The novelty of the work is not clear. The effect of the reduction temperature on the electrocatalytic and physicochemical properties of GO has been widely examined in the past decade, including several works concerning the oxygen reduction reaction. Moreover, recently, investigations have been extended to advanced GO-based samples, indicating the exhaustion of the pure GO examination. What is exactly the novel aspect of the current work relative to similar works previously reported? Is it the examination of various quantities of alumina boats and GO quantities during thermal treatment? Generally, this novelty should be highlighted in the introduction by providing the relevant references and thoroughly discussing them compared to your research. By the way, half of the introduction is devoted to general information about graphene and GO, which has already been reported in several manuscripts. It would be sufficient to focus specifically on the beneficial effect of GO on ORR and on highlighting the novelty. Finally, it is not clear for which applications that include ORR the as-prepared catalyst is suggested.

Response 2: Thank you very much for your valuable comments. The main novelty of the work is the direct use of graphene as a 4-electron transfer electrocatalyst for ORR. In general, most of the reported carbon samples can show 2-electron transfer without the introduction of heteroatom or transition metals, or Nobel metals. Our synthesized GO also displayed 2-electron transfer pathways during ORR. At the same time, the pyrolysis of GO at 900 °C with the use of 100 mg in 1 or 2 boats would facilitate almost the direct 4-electron transfer. The 4-electron transfer ORR activity of graphene was also confirmed further by preparing various graphenes with similar approaches and finding that the use of an optimum amount of GO during the pyrolysis can facilitate better electrocatalytic ORR activity (Page 3, Line numbers 115-123).

3) A literature review paragraph should be included in the introduction section to highlight the different ways for ORR (last five years). See for example works of prof. Shuqin Song and prof. Zhenxing Liang groups.

Response 3: We have updated the literature survey with recent papers in the revised manuscript. We have also added some papers of Prof. Shuqin Song and Prof. Zhenxing Liang groups. 

4) More of the half references are older than 5-6 years. They should be updated.

Response 4: Thank you very much for your valuable suggestion. We have updated the recently published articles in the revised manuscript according to your suggestion which are mentioned in red color.

5) The nature and the role of the catalyst active sites for ORR should be better and clearly pointed out. The authors should specifically explain the role by connecting the physicochemical characterization results with the photocatalytic activity of the prepared catalyst.

Response 5: Thank you very much for your valuable suggestion. We are sorry, we don’t find the exact reason for the better ORR activity of our prepared graphene as compared with the published works. Because, most of the reported graphene electrocatalyst are modified with heteroatoms, transition metals, and Nobel metal to achieve 4 electron transfer. Meanwhile, in our prepared graphene displayed the n value of 3.76. We think, the better ORR activity of the optimized graphenes may be due to the presence of higher valence orbital energies of the ORR active atom, which increases the adsorption of OOH* and OH* intermediates (Page 13, Line numbers 472-474).

6) In the abstract and introduction sections, numerical ORR activity results should be highlighted. More information about the main findings should be reported in Abstract section.

Response 6: Thank you very much for your valuable suggestion. We have updated the abstract in the revised manuscript according to your suggestion (Page 1, Line numbers 27-36).

7) The phrase “platinum group-free metal graphene” is not adequate. “Metal-free” would be more sufficient, since the as-prepared graphene is pure. Moreover, the notation X and Y for 50 and 100 mg of GO is complicated. You could note the samples as G50 and G100 instead of GX and GY.

Response 7: Thank you very much for your valuable suggestion. We have updated the title of the manuscript and sample names in the revised manuscript according to your suggestion (Page 1, Line number 2).

8) It is argued that the annealing at 900 oC produces loose and thin graphene sheets that they could assist the catalyst ink dispersion on the working electrode. This argument does not seem correct. How is the microstructure correlated with the ink dispersion? Presumably you were trying to refer to the electrochemical active surface area.  

Response 8: Thank you very much for your valuable question. We think the explanation of the catalyst ink for G50 and G100 was not clearly understood. We have rewritten the sentence in the revised manuscript. The use of 50 mg of GO in 1 to 3 alumina boats would produce almost crystalline graphene particles (G50-1b to 3B) after pyrolysis at 900 °C. The catalyst ink prepared from these samples is dispersed during ultrasonication and loaded on the glassy carbon substrate before analyzing the ORR. At the same time, the pyrolysis performed for the samples with 100 mg of GO in each boat may have shown the formation of lightweight graphene sheets (G100-1B and 2B) as compared to the use of 50 mg of GO (crystalline graphene) due to the better exfoliation and thermal expansion of graphene sheet while introducing a larger quantity of GO in a single boat. For comparison, we also checked this behaviour by further increasing the GO amount in a single boat which induces the faster expansion of GO sheets even the temperature reaching to above 200 to 250 °C and spread in the glass tube. This leads to loose of some samples during collection. To avoid sample loss, we used the optimum amounts of GO (100 mg) samples in a boat for better exfoliation and to avoid the loss of more graphene samples after pyrolysis at 900 °C. The catalyst ink prepared with G100 showed well dispersed like Pt/C ink as compared with the ink prepared with G50. Owing to changes in the expansion and surface morphological differences between G50 and G100, we obtained better electrocatalytic ORR activity for G100 as compared with G50. These results suggest that the use of the optimum amount of GO and thermal annealing temperature plays vital roles to get graphene sheets without any loss of samples. We hope the electrochemically active surface area may be better for G100 than G50 (Page 6, Line numbers 255-267 and Page 7, Line number 268-273).  

9) The effect of the alumina boats and GO quantity on the physicochemical features of the as-prepared samples are not explained but only demonstrated in the corresponding characterizations.

Response 9: Thank you very much. We provided a more detailed explanation of the effect of the alumina boats and GO quantity in the revised manuscript (Page 6, Line numbers 255-267, and Page 7, Line number 268-273).

10) It is argued that RGOY presents ORR activity in N2 saturated solution. This argument is incorrect. The obtained CV demonstrates the surficial oxide formation/reduction, either because of the reorganization of the catalyst surface or interaction with the electrolyte ions. Moreover, such a CV figure could indicate a not well-organized catalytic surface area, which could also be argued according to the large hysteresis of the CV in the O2-saturated electrolyte. If this is the case, a conditioning procedure is required before conducting ORR measurements for reliable and reproducible results. Indicatively, the kinetic region seems to change slightly with the rpm alternation for the RGOY sample (Figure 9), which should not be the case. Probably the measurements were not reproducible.

Response 10: We rechecked RGOY few times to confirm the electrocatalytic ORR acticity which showed almost similar results. To avoid much confusion, we removed the RGOY contents in the revised manuscript.

11) For reliable Koutecky-Levich analysis, the ORR LSV curves should be corrected for the ohmic solution resistance and the capacitive (background) currents.

Response 11: Thank you very much. We plotted the LSV graph after zero-correction. We will consider your comments in our future works.  

12) The ORR results analysis should be enhanced, e.g., calculate kinetic currents, exchange current densities, provide durability results, etc. 

Response 12: Thank you very much for your valuable comments. This is our initial experimental result by the use of the prepared graphene for ORR. Unfortunately, we could not perform the further experiment in this work. So, we revised the manuscript based on the initial experimental study. We deeply consider your points for our future experimental studies related to ORR.    

13) According to Table S1 and other reported works, the achieved ORR results are not exceptional. Based on this fact, you should not exaggerate about the contribution of the current work.   

Response 13: We agree with your point. Most of the reported graphene-based materials are modified with various modifiers such as heteroatom, transition metals, and Nobel metals. It is well-known that after some modifications graphene can show better ORR activity than pristine graphene. At the same time, our preliminary experimental results showed that the prepared graphene can show better ORR activity. 

14) The manuscript is not well-organized. The main ORR results should be transferred from the “Discussion” section to the “Results” section. The “Discussion” section should be exclusively devoted on correlating the electrocatalytic behavior of the prepared samples with the physicochemical results, while highlighting the effect of the alumina boats and GO quantities during the thermal treatment. 

Response 14: We merged the results and discussion sections into one section in the revised manuscript (Page 5, Line number 238).

Reviewer 2 Report

In this paper, the reduction of graphene oxide by the modified Hummer’s method, the characterization of the resulting graphene and their oxygen reduction reaction activity are described. This paper is very interesting and meaningful for electrocatalytic oxygen reduction.  I recommend that this paper is published after minor revisions according to a following comments.

Comments:

1.     (L223-503) It is better to combine "3. Results" and "4. Discussion" into "3. Results and Discussion".

2.     (L211) References 19 and 21 do not explain Equation 1.

3.      ORR activity differs greatly between GX1B and GY1B, but what are the structural differences between them?

4.      It is better to explain the three types of reduction, 2-electron reduction, 4-electron reduction, and reduction out of the 2 and 4 electron transfer, by showing reaction formulas. The authors should also explain why there are such differences between the samples.

 5. Others

L42: produce graphene oxide (GO), ---> produce GO,

L42-43: the reduction of graphene oxide ---> the reduction of GO

L66: The reduction of graphene oxide into ---> The reduction of GO into

L80: N-doped RGO and ---> N-doped rGO and

L123: Graphene oxide was ---> GO was

L125: graphite flask ---> graphite flake

L143: deionized ---> deionized

L148-149: In general, graphene oxide has ---> In general, GO has

L170: as reduced graphene oxide (RGOY 3B). ---> as rGO (named RGOY 3B).

L182-183: in graphene and graphene oxide ---> in graphene and GO

L214: equation 1-4 ---> equation 2-4

L216: Co2 ---> C0,   (D0)2/3w1/2n-1/6 ---> (D0)2/3n-1/6

L221: 1.9 x 10-5 cm s-1 ---> 1.9 x 10-5 cm2 s-1,  0.01 cm-2 s-1 ---> 0.01 cm2 s-1

L246, 292, 314, 401: and (g) GY 3B. ---> and (g) RGOY 3B.

L283: (Fig. 4B(b and c)) --->  (Fig. 4B(e and f))

L284: (Fig. 4B(d)) ---> (Fig. 4B(g))

L284: and RGO at 250 °C ---> and rGO at 250 °C

L296, 313, 400, 424, 455: Go ---> GO

L305: absorbed ---> adsorbed

L309: the formation of RGO at 250 °C ---> the formation of rGO at 250 °C

L311: RGO. ---> rGO.

L338: restored [28]. ---> restored [27].

L372: of graphene oxide (GO), ---> of GO,

L387: at the N2 and O2 saturated  --->  at the N2 saturated

L424, 455: and (G) GY 3B ---> and (G) RGOY 3B

L431: stopped GY 3B ---> stopped RGOY 3B

L440: graph of RGO 3B ---> graph of RGOY 3B

L458: K-L plot ---> Koutecky-Levich (K-L)plot

L458: the most continent method ---> the most common method

L459-460: Koutecky-Levich plots ---> K-L plots

L468: each sheet of RGO layers. ---> each sheet of rGO layers.

L477: Koutecky-Levich plot ---> K-L plot

L493: with GO and RGO ---> with GO and rGO

L506: graphene and RGO ---> graphene and rGO

L517: GO, graphenes, and RGO. ---> GO, graphenes, and rGO.

L519: with GO and RGO. ---> with GO and rGO.

Axis titles

Figures 8 and 9: E/V vs. RHE ---> E (V vs. RHE),  J/mA cm-2 ---> J (mA cm-2)

Figure 10: w-0.5/rpm-0.5 ---> w-0.5 (rpm-0.5), J-1mA-1cm-2 ---> J-1 (mA-1cm2)

Author Response

Answers to the reviewer's comments

Reviewer 2

In this paper, the reduction of graphene oxide by the modified Hummer’s method, the characterization of the resulting graphene and their oxygen reduction reaction activity are described. This paper is very interesting and meaningful for electrocatalytic oxygen reduction.  I recommend that this paper is published after minor revisions according to a following comments.

Thank you very much for reviewing our manuscript also for your valuable comments to improve the quality of our manuscript. We carefully revised all your comments and inputted the answer to all your queries point-by-points and mentioned the changes in the revised manuscript in red colour.      

Comments:

  1. (L223-503) It is better to combine "3. Results" and "4. Discussion" into "3. Results and Discussion".

Response 1: Thank you very much. We combined the "3. Results" and "4. Discussion" into one section "3. Results and Discussion" in the revised manuscript (Page 5, Line number 238).

  1. (L211) References 19 and 21 do not explain Equation 1.

Response 2: We corrected the mistakes in the references and mentioned the revised references in red color.

  1. ORR activity differs greatly between GX1B and GY1B, but what are the structural differences between them?

Response 3: The surface morphology between G50-1B and G100-1B greatly differs when increasing the GO amount to pyrolysis temperature at 900 °C. Moreover, the particulate structure has some differences. G50-1B has crystalline particles after pyrolysis, whereas, G100-1B have produce lightweight graphene sheets.    

  1. It is better to explain the three types of reduction, 2-electron reduction, 4-electron reduction, and reduction out of the 2 and 4 electron transfer, by showing reaction formulas. The authors should also explain why there are such differences between the samples.

Response 4: Thank you very much for your comments. Commonly, around 2 electron transfers occur for the pristine carbon sample and 4 electron transfers occur for the modified samples. In some cases, the carbon samples also have multi-electron transfer pathways based on their surface nature. We don’t know the exact reason for the multi-electron transfer. To avoid confusion, we removed rGO results from the revised manuscript and updated the full manuscript.  

  1. Others

L42: produce graphene oxide (GO), ---> produce GO,

L42-43: the reduction of graphene oxide ---> the reduction of GO

L66: The reduction of graphene oxide into ---> The reduction of GO into

L80: N-doped RGO and ---> N-doped rGO and

L123: Graphene oxide was ---> GO was

L125: graphite flask ---> graphite flake

L143: deionized ---> deionized

L148-149: In general, graphene oxide has ---> In general, GO has

L170: as reduced graphene oxide (RGOY 3B). ---> as rGO (named RGOY 3B).

L182-183: in graphene and graphene oxide ---> in graphene and GO

L214: equation 1-4 ---> equation 2-4

L216: Co2 ---> C0, (D0)2/3w1/2n-1/6 ---> (D0)2/3n-1/6

L221: 1.9 x 10-5 cm s-1 ---> 1.9 x 10-5 cm2 s-1, 0.01 cm-2 s-1 ---> 0.01 cm2 s-1

L246, 292, 314, 401: and (g) GY 3B. ---> and (g) RGOY 3B.

L283: (Fig. 4B(b and c)) --->  (Fig. 4B(e and f))

L284: (Fig. 4B(d)) ---> (Fig. 4B(g))

L284: and RGO at 250 °C ---> and rGO at 250 °C

L296, 313, 400, 424, 455: Go ---> GO

L305: absorbed ---> adsorbed

L309: the formation of RGO at 250 °C ---> the formation of rGO at 250 °C

L311: RGO. ---> rGO.

L338: restored [28]. ---> restored [27].

L372: of graphene oxide (GO), ---> of GO,

L387: at the N2 and O2 saturated  --->  at the N2 saturated

L424, 455: and (G) GY 3B ---> and (G) RGOY 3B

L431: stopped GY 3B ---> stopped RGOY 3B

L440: graph of RGO 3B ---> graph of RGOY 3B

L458: K-L plot ---> Koutecky-Levich (K-L)plot

L458: the most continent method ---> the most common method

L459-460: Koutecky-Levich plots ---> K-L plots

L468: each sheet of RGO layers. ---> each sheet of rGO layers.

L477: Koutecky-Levich plot ---> K-L plot

L493: with GO and RGO ---> with GO and rGO

L506: graphene and RGO ---> graphene and rGO

L517: GO, graphenes, and RGO. ---> GO, graphenes, and rGO.

L519: with GO and RGO. ---> with GO and rGO.

Response 5: Thank you very much for indicating the mistakes. We have corrected all the mistakes in the revised manuscript according to your comments.

  1. Axis titles

Figures 8 and 9: E/V

  1. RHE ---> E (V vs. RHE), J/mA cm-2---> J (mA cm-2)

Response 6: Thank you very much. We have corrected the axis units of electrochemical activities according to your suggestions.

  1. Figure 10: w-0.5/rpm-0.5---> w-0.5 (rpm-0.5), J-1mA-1cm-2 ---> J-1 (mA-1cm2)

Response 7: Thank you very much. We have corrected the axis units of electrochemical activities according to your suggestions.

Reviewer 3 Report

Dear Authors

The manuscript is focused on the investigation of prepared graphene from graphene oxide (GO) by pyrolysis method under a nitrogen atmosphere at 900 ºC have shown better ORR activity in aqueous 0.1M potassium hydroxide solution electrolyte as compared with the electrocatalytic activity of pristine GO.

The manuscript presented concerns an interesting and actual subject.

The following suggestion and comments should be taken:

1. The overall English needs to be improved. Please seek guidance from a native English speaker if possible ("the" "a", commas, plural form and others could be corrected).

2. The authors could insert more numerical data into the Abstract for enhancement of the manuscript.

3. Please better explain the novelty of your work.

4. The introduction section needs enhancement few sentences about

(1) The application of graphene and its composites in oxygen reduction electrocatalysis: a perspective and review of recent progress Energy Environ. Sci., 2016, 9, 357-390 https://doi.org/10.1039/C5EE02474A.

(2) High surface area micro-mesoporous graphene for electrochemical applications. Sci Rep 11, 22054 (2021). https://doi.org/10.1038/s41598-021-01154-0

(3) A review of oxygen reduction mechanisms for metal-free carbon-based electrocatalysts npj Comput Mater 5, 78 (2019). https://doi.org/10.1038/s41524-019-0210-3

5. Due to the number of acronyms used in this manuscript, authors must include a list of abbreviations.

6. Could the authors include the standard deviation of the used methods?

7. Raman spectroscopy. Could authors add a table with an intensity ratio and add more comments about the quality of materials?

8. Figure 6. Please correct the X-axis descriptions.

9. Figure 10. Please correct this image for better quality (the inscriptions).

10. The authors are suggested to discuss the validity and reproducibility of the ORR performance values.

11. The authors are advised to detail the ORR mechanism of the samples. Please see the article J Nanopart Res 25, 28 (2023). https://doi.org/10.1007/s11051-023-05671-z and others.

12. ECSA (electrochemically active surface area) is a useful parameter to evaluate electrochemical electrodes (J. Mater. Chem. A, 2020, 8, 13459). Have the authors measured it for the samples? Can you comment this with relevant refs?

13. It will be good to introduce the number of electrons of commercial electrode Pt/C for comparison.

14. Why author choose these systems for the study and temperature 900, not 700 or 800 or 1000 oC? Please explain.

15. Please describe the fitting procedure for XPS spectra.

16. Authors are suggested to describe some potential applications for samples in conclusions.

Author Response

Answers to the reviewer's comments

Reviewer 3

The manuscript is focused on the investigation of prepared graphene from graphene oxide (GO) by pyrolysis method under a nitrogen atmosphere at 900 ºC have shown better ORR activity in aqueous 0.1M potassium hydroxide solution electrolyte as compared with the electrocatalytic activity of pristine GO.

The manuscript presented concerns an interesting and actual subject.

The following suggestion and comments should be taken:

Thank you very much for reviewing our manuscript also for your valuable comments to improve the quality of our manuscript. We carefully revised all your comments and inputted the answer to all your queries point-by-points and mentioned the changes in the revised manuscript in red colour.      

  1. The overall English needs to be improved. Please seek guidance from a native English speaker if possible ("the" "a", commas, plural form and others could be corrected).

Response 1: We have corrected the grammatical errors in the revised manuscript.

  1. The authors could insert more numerical data into the Abstract for enhancement of the manuscript.

Response 2: We have added the numerical values in the abstract of the revised manuscript (Page 1, Line numbers 27-36).

  1. Please better explain the novelty of your work.

Response 3: Thank you very much for your valuable comments. The main novelty of the work is the direct use of graphene as a 4-electron transfer electrocatalyst for ORR. In general, most of the reported carbon samples can show 2-electron transfer without the introduction of heteroatom or transition metals or Nobel metals. Our synthesized GO also displayed 2-electron transfer pathways during ORR. At the same time, the pyrolysis of GO at 900 °C with the use 100 mg in a 1 or 2 boats would facilitate almost the direct 4-electron transfer. This is the main novelty of the work. To confirm the 4-electron transfer ORR activity of graphene, we have checked the electrocatalytic activity under various GO and graphene samples prepared by altering the GO ratio (Page 3, Line numbers 115-123).   

  1. The introduction section needs enhancement few sentences about

(1) The application of graphene and its composites in oxygen reduction electrocatalysis: a perspective and review of recent progress Energy Environ. Sci., 2016, 9, 357-390 https://doi.org/10.1039/C5EE02474A.

(2) High surface area micro-mesoporous graphene for electrochemical applications. Sci Rep 11, 22054 (2021). https://doi.org/10.1038/s41598-021-01154-0

(3) A review of oxygen reduction mechanisms for metal-free carbon-based electrocatalysts npj Comput Mater 5, 78 (2019). https://doi.org/10.1038/s41524-019-0210-3

Response 4: We have provided more detailed introduction and also added the suggested references as well as updated with more recent literatures in the revised manuscript (Page 1, Line numbers 40-47).

  1. Due to the number of acronyms used in this manuscript, authors must include a list of abbreviations.

Response 5: Thank you very much for your valuable suggestion. We have added the abbreviations in the revised manuscript (Page 17, Line numbers 564-573).

GO: graphene oxide, Eonset: onset potential, E1/2: half-wave potential, JL: limiting current density, n: electron transfer number, ORR: oxygen reduction reaction, rGO: reduced graphene oxide, G50-1B, 2B, and 3B: 50 mg of GO in 1, 2, and 3 alumina boats pyrolyzed at 900 °C, G100-1B and 2B: 100 mg of GO in 1 and 2 alumina boats pyrolyzed at 900 °C, SCCM: standard cubic centimeter per minute, CV: cyclic voltammetry, LSV: linear sweep voltammetry, RRDE: rotating ring-disk electrode, SCE: saturated calomel electrode, GCE: glassy carbon electrode, RHE: reversible hydrogen electrode, FESEM: Field Emission Scanning Electron Microscopy, FETEM: Field Emission Transmission Electron Microscopy, XRD: X-ray diffraction, FTIR, Fourier Infra-red Spectroscopy, XPS: X-ray Photoelectron Spectroscopy, rpm: rotation per minute, (K-L) plot: Koutecky-Levich plot, Pt/C: platinum/carbon, NG: N-doped graphene.

  1. Could the authors include the standard deviation of the used methods?

Response 6: We are sorry for not including the standard deviation of the used methods. We performed electrochemical activity 2 times for each sample and the samples showed almost similar results. We plotted the graphs from the obtained data.

  1. Raman spectroscopy. Could authors add a table with an intensity ratio and add more comments about the quality of materials?

Response 7: We are sorry for not providing more information regarding Raman spectroscopy. We will consider your valuable point and add such data in our future submissions.

  1. Figure 6. Please correct the X-axis descriptions.

Response 8: We corrected the axis descriptions in the revised manuscript.

  1. Figure 10. Please correct this image for better quality (the inscriptions).

Response 9: We have updated Fig. 8 to 10 in the revised manuscript.

  1. The authors are suggested to discuss the validity and reproducibility of the ORR performance values.

Response 10: We performed electrochemical activity 2 times of each samples and the samples showed almost similar results. We plotted the graphs from the obtained datas. We hope, the graphene electrocatalyst is stable and easily reproducible.

  1. The authors are advised to detail the ORR mechanism of the samples. Please see the article J Nanopart Res25, 28 (2023). https://doi.org/10.1007/s11051-023-05671-z and others.

Response 11: We are sorry for not providing the detailed mechanism due to a lack of information. Because the ORR activities mainly take place at the surface. Even though, the carbonized samples not having any functional groups we found better electrocatalytic activity in our prepared graphene. Instead, we have updated the mentioned references in the revised manuscript.

  1. ECSA (electrochemically active surface area) is a useful parameter to evaluate electrochemical electrodes (J. Mater. Chem. A, 2020, 8, 13459). Have the authors measured it for the samples? Can you comment this with relevant refs?

Response 12: We are sorry, we didn’t analyse the ECSA parameter. We will consider your valuable comments and do them in our future work on this study.

  1. It will be good to introduce the number of electrons of commercial electrode Pt/C for comparison.

Response 13: The calculated n value for Pt/C is 3.71 which is mentioned in the revised manuscript.

  1. Why author choose these systems for the study and temperature 900, not 700 or 800 or 1000 oC? Please explain.

Response 14: We are sorry for not performing the pyrolysis temperature at various temperatures due to the easier expansion of GO above 100 mg in a single boat and at above 200-250 °C. In this work, our preliminary experiment mainly focuses on the preparation of graphene in larger quantities in a single batch of pyrolysis. So, we fixed a particular amount and performed the pyrolysis at 900 °C. This is our initial attempt to try the prepared graphenes for the ORR electrocatalyst. Above 800-900 °C, pyrolyzed graphene can have excellent conductivity and other various properties. So, we fixed the pyrolysis temperature at 900 °C and performed further experiments. We also consider pyrolysis of various other temperatures in the future and study the electrocatalytic activity.

  1. Please describe the fitting procedure for XPS spectra.

Response 15: We provided the fitting procedure for XPS spectra in the revised manuscript. The CASA XPS software was used to perform peak-fitting analyses and quantitative observations.

  1. Authors are suggested to describe some potential applications for samples in conclusions.

Response 16: We added some potential applications of the prepared electrocatalysts in the revised manuscript.

Round 2

Reviewer 1 Report

The authors seriously considered my suggestions and appropriately impeoved their manuscript. I think now it could be accepted for publication to Nanomaterials.

Just a last thing: For aesthetic reasons, please make Figs 4-6 the same size as the others.

Reviewer 3 Report

Dear Authors

The authors have addressed all comments and the manuscript can be published as is.